# A MD Simulation Prediction for Regulation of *N*-Terminal Modification on Binding of CD47 to CD172a in a Force-Dependent Manner

**DOI:** 10.3390/molecules28104224

**Published:** 2023-05-22

**Authors:** Yang Zhao, Liping Fang, Pei Guo, Ying Fang, Jianhua Wu

**Affiliations:** Institute of Biomechanics, School of Biology and Biological Engineering, South China University of Technology, Guangzhou 510006, China

**Keywords:** CD47, CD172a, post-translated modification, molecular dynamics simulation, structure–function relation, mechano-chemical regulation

## Abstract

Cancer cells can evade immune surveillance through binding of its transmembrane receptor CD47 to CD172a on myeloid cells. CD47 is recognized as a promising immune checkpoint for cancer immunotherapy inhibiting macrophage phagocytosis. *N*-terminal post-translated modification (PTM) via glutaminyl cyclase is a landmark event in CD47 function maturation, but the molecular mechanism underlying the mechano-chemical regulation of the modification on CD47/CD172a remains unclear. Here, we performed so-called “ramp-clamp” steered molecular dynamics (SMD) simulations, and found that the *N*-terminal PTM enhanced interaction of CD172a with CD47 by inducing a dynamics-driven contraction of the binding pocket of the bound CD172a, an additional constraint on CYS15 on CD47 significantly improved the tensile strength of the complex with or without PTM, and a catch bond phenomenon would occur in complex dissociation under tensile force of 25 pN in a PTM-independent manner too. The residues GLN52 and SER66 on CD172a reinforced the H-bonding with their partners on CD47 in responding to PTM, while ARG69 on CD172 with its partner on CD47 might be crucial in the structural stability of the complex. This work might serve as molecular basis for the PTM-induced function improvement of CD47, should be helpful for deeply understanding CD47-relevant immune response and cancer development, and provides a novel insight in developing of new strategies of immunotherapy targeting this molecule interaction.

## 1. Introduction

The transmembrane protein CD47, also known as an integrin-associated protein (IAP) and served as a self-recognition marker [1], is ubiquitously expressed in human cells, various hematologic and solid tumors [2], and involved in cell apoptosis, proliferation, adhesion, migration and tumor development, as well as immune responses [3,4,5,6,7]. Binding to CD172a (the signal regulatory protein α, SIRPα) on macrophage, CD47 on tumor cells can send the inhibitory signal “do not eat me” to macrophages so as to maintain the immune tolerance of own cells, prevent tumor cells from being phagocytized, and inhibit tumor antigen presentation [8,9]. The overexpression of CD47 has become a key strategy of the tumor cell in evading macrophage-mediated phagocytosis [10,11]. Thus, CD47 has emerged as a promising checkpoint molecule for anti-tumor therapy [12,13].

CD47 is a 50 kDa membrane receptor consisting of an extracellular *N*-terminal IgV-like domain, five transmembrane helices, and a short C-terminal intracellular tail [14,15]. The extracellular IgV-like domain contains five N-glycosylation sites, while there exist four kinds of spliceosomes in the intracellular regions of 4–36 amino acids (Figure 1) [15,16,17]. The distribution and expression of these spliceosomes are different in vivo; spliceosome 2 of sixteen amino acids is the most widely expressed and distributed on the surfaces of almost all hematopoietic cells, endothelial cells, and epithelial cells [16]. Crystallographic studies show very little ligand-induced change in CD47 conformation [17]. The disulfide bond between CYS15 in the extracellular domain and CYS245 in the transmembrane domain of CD47 is susceptible to the redox potential change, and leads CD47 to take an orientation in favor of ligation binding and the subsequent signal transduction [14,17,18]. CD172a is a crucial ligand of CD47, belongs to the immunoglobular superfamily, and has an extracellular region of three Ig-like domains, in which only the first Ig-like domain participates in binding with CD47 (Figure 1), and an intracellular domain of two immunoreceptor tyrosine-based inhibitory motifs (ITIMs) [19]. CD47 promotes the tyrosine phosphorylation of CD172a ITIMs, causing recruitment of Src homology region 2 domain-containing phosphatase-1 and 2 (SHP-1 and 2) and inhibiting phagocytosis [20,21].

Passing through Golgi apparatus, CD47 will be catalyzed by glutaminyl cyclase, which converts a glutamine at the *N*-terminus to a pyroglutamic acid (PCA) and is dominant for CD47 signaling [22,23]. This post-translational modification (PTM) significantly improves CD47’s affinity with CD172a, possibly by increasing protein resistance to enzymatic degradation [24]. However, the dynamics mechanism for regulation of the *N*-terminal modification on CD47 affinity with CD172a is unclear, even though the solved crystal structure of CD47/CD172a complex has shown two hydrogen bonds between SER66 on CD172a and the pyroglutamic acid, the *N*-terminal residue of CD47 [17]. Additionally, several lines of evidence indicate biochemical regulation in the interaction of CD47 with CD172a. Cytokine stimulation in HLH patients reduces CD47 expression [25], interleukin 4 (IL-4), IL-7, and IL-13 influence CD47 expression on Sezary cells [26], and the expression level of CD47 in cancer cells is regulated by miR-133a, MYC oncogene, and the ERK signaling pathway [27,28,29]. A transformation from the “do not eat me” signal to “eat me” one is believed to relevant to a conformation change of CD47 for the experimental aging red blood cells [30].

Numerous studies have shown mechano-sensing properties of immunoreceptors [31]. Periodic mechanical force stimulation of human chondrocytes can elicit a CD47-dependent change in the membrane potential, showing an actor of CD47 as a force sensor [32]. CD47-antibody-induced apoptosis occurs in tumor cells on hard substrates rather than soft gels [33], suggesting a requirement of substrate stiffness for CD47-modulated cancer development. The Protein Linking Integrin-Associated Complexes (PLICs) can act as a linker of the CD47 cytoplasmic tail and the cell skeleton [34], transmitting transmembrane mechano-chemical signaling. Additionally, interaction of CD47 with either thrombospondin and/or CD172a can promote the integrin-dependent recruitment of T cells towards endothelial cells [35,36]. These studies mentioned above do suggest a possible deep involvement of mechano-microenvironment in CD47-relevant cellular physio-pathological processes, but there is less knowledge of the mechanism of mechanical regulation on interaction of CD47 with CD172a.

Herein, we investigate the effects of PTM and tensile force on the interaction of CD47 with CD172a through molecular dynamics (MD) simulations with two molecular systems using CD172a bound to CD47 with or without PTM. Our results state that the PTM-induced local flexibility change in CD172a might be responsible for the upregulation of CD47 affinity to CD172a, and, the *N*-terminal PTM and tensile force may improve CD47 affinity to CD172a by stabilizing the binding pocket of CD172a. The present study provides a novel insight into the dynamics mechanism and molecular basis for CD47/CD172a interaction under tensile force with or without PTM. This should assist with our understanding of CD47-involved cellular immune response and tumor development and should be useful for developing novel therapeutic strategies targeting CD47 in cancers.

## 2. Results

### 2.1. N-Terminal Post-Translational Modification Improved CD47 Affinity to CD172a

To assess the role of post- translational modification (PTM) on binding of CD47 with CD172a at the atom level, free MD simulation of 40 ns was performed thrice for each of two different equilibrated CD172a/CD47 complexes, which took its wild-type (WT) or GLN-type (GT), respectively (Figure 1) (Materials and Methods). The WT complex had a PTM on *N*-terminus of CD47 and lacked the GT one.

We found from the time courses of Cα-RMSD (Figure 2A) that the Cα-RMSD pattern of WT complex has less difference from that of GT complex, while Cα-RMSD fluctuated slightly, increased slowly with time, and then reached a plateau of about 1.75 Å, while it varied within a range of 1 Å throughout the simulation time no matter whether CD47 was WT or GT, showing a high thermal stability of CD172a/CD47 complex in a PTM-independent manner. N_HB_, the mean number of hydrogen bonds across the interfacial surface, was calculated to be 9.8 ± 0.27 for the WT complex and 8.7 ± 0.16 for the GT complex (Figure 2B), exhibiting a PTM-enhanced H-bonding between CD172a and CD47. As a result, a lower dissociation probability *f*_D_ (= (6.77 ± 7.48) × 10^−7^) was predicted for the WT complex in comparison with *f*_D_ (= (1.30 ± 0.63) ×10^−5^) of the GT complex (Figure 2C). This PTM-induced upregulation of CD47 affinity to CD172a was reflected by the PTM-induced increase in the mean buried SASA and decrease in both the mean binding energy and the mean Rgyr (Figure 2D–F). The present data showed that the mean binding energies (E_B_) were -553.7 ± 35.13 and −520.4 ± 16.09 kcal/mol, the mean Buried-SASA values were 9.9 ± 0.01 and 9.1 ± 0.16 nm^2^, and the mean rotation radius (Rgyr) were 19.9 ± 0.03 and 20.2 ± 0.07 Å for the WT and GT complexes, respectively (Figure 2D–F). These data had a PTM-upregulated affinity of CD172a to CD47. This result matched well with a previous report that the *N*-terminal modification of CD47 could enhance the binding ability of CD47 to CD172a [23].

### 2.2. PTM Enhanced Interfacial H-Bonding and Induced Contraction of Binding Pocket of Bound CD172a

To investigate the structural basis for the PTM-enhanced CD47 affinity with CD172a, we further examined the effect of CD47 *N*-terminal PTM on the complex conformation through free MD simulations. Three geometrical characteristics, consisting of the distance (*H*) between the centroids of CD and DE loops in CD172 (Figure 3A), the angle (*α*) between the C strand of CD172a and the straight line extending from the Cα atom of GLN52 to the Cα atom of LYS53 in CD172a (Figure 3B), and the angle (*θ*) between the G strand in CD47 and the C strand in CD172a (Figure 3C), were used to quantify conformational evolution under thermal excitation.

We found that high conformational conservation lay in the bound CD47 with or without PTM, while a significant PTM-induced conformation change occurred in the binding pocket of the bound CD172a. This PTM-induced allostery of CD172a was demonstrated by the PTM-induced decreases in *H* and *α* (Figure 3D,E). The values of H and *α* were 16.1 ± 0.4 Å and 69.8° ± 2.7° for the WT complex and 17.4 ± 0.3 Å and 90.0° ± 2.7° for the GT complex, respectively, while a reduction of *θ* from 95.4° ± 1.3° to 89.5° ± 0.4° occurred (Figure 3F). This allostery stated a PTM-induced contraction of the binding pocket of the bound CD172a, brought the CD loop closer to the BC and DE loops for the bound CD172a (Figure 1 and Figure 3A,B), and led to closer contact between CD47 and CD172a by causing a roll of CD172a on CD47 (Figure 3C,F), saying that contraction of the binding pocket of CD172a was involved in PTM-induced enhancement of CD47 affinity to CD172a. Additionally, the mean RMSF pattern of CD172a showed that flexibilities of the CD loop (the 50th–60th residues) in the WT complex were smaller than that of the GT complex (Figure 3G), suggesting a more stable CD172a binding pocket or a stronger CD47-related constraint on CD172a in the WT complex instead of the GT one.

Meanwhile, we detected twelve hydrogen bonds in the binding site from thrice 40ns free MD simulations (Table 1, Figure 1C–E). Of all these bonds, seven were enhanced significantly by the *N*-terminal PTM of CD47, and the involved residues were ASP57, GLU97, GLU100, GLU104, GLU106, GLU100, and PCA1 on CD47, as well as their respective partners, such as GLN52, LYS53, SER66, ARG69, LYS96, and SER98 in CD172a (Table 1). They should play a crucial role in the PTM-induced enhancement of CD47 affinity with CD172a, although PTM showed a stronger H-bond (with occupancy of 77%) between ASP51 on CD47 and ARG95 on CD172a very weak (Table 1). The five key residues on CD172a were located in the CD, DE and FG loop (Figure 1C–E). Additionally, three GLN1-involved H-bonds were weak, with survival ratios smaller than 0.25, and would vanish completely with the formation of a moderate H-bond (with a survival ratio of 0.52) between PCA1 on CD47 *N*-terminus and SER66 on the DE loop of CD172a, hinting that a reinforcement signal for CD47 binding with CD172 was triggered through converting a glutamine at *N*-terminus of CD47 to a pyroglutamic acid. The H-bond between GLU35 on CD47 and ARG69 on CD172a was strong, with high occupancy larger than 80% no matter whether PTM occurred at the CD47 terminal or did not, indicating that this residue pair should be crucial for binding of CD47 to CD172a.

### 2.3. Mechanical Constraint on CYS15 of CD47, Instead of N-Terminal PTM, Might Improve Tensile Strength of CD172a/CD47 Complex

A stable transmembrane signal transduction through the CD47/CD172a axis required better mechanical strength for the CD47/CD172a complex, especially in the mechano-microenvironment. Herein, we performed “force-ramp” SMD simulation over 10ns thrice with pulling velocity of 5 Å/ns for each system to investigate the mechanical strength of the mechano-constrained complex with or without PTM. Two loading modes, mode 1 and 2, were applied in SMD simulations, no matter whether the bound CD47 was one with *N*-terminal PTM or not. In each loading mode, the Cα atom of CD172a C-terminal (ALA115) was selected as the steered one; and in mode 2, two Cα-atoms of CD47, one on C-terminal (VAL115) and another on CYS15, were fixed, but in mode 1, only the C-terminal Cα-atom of CD47 was fixed (Figure 4C).

The force–time curve showed that the tension rose rapidly to its peak (the rupture force) of about 530 pN in mode 2 or about 420 pN in mode 1 (Figure 4A,D) in either PTM presence or absence, exhibiting a strong resistance to the pull-induced CD172a dissociation from CD47 in a manner independent of PTM and the loading modes. However, the present data revealed that the mechanical constraint on CYS15 of CD47 improved the tensile strength of the complex and did not improve the CD47 *N*-terminal PTM. It was believed that a disulfide bond could form between CYS15 in the extracellular domain and CYS245 in transmembrane domains of CD47 [17], meaning that the formation of the disulfide bond provided a mechanical intramolecular constraint on CD47, and made CD47 anchor tightly to the cellular membrane. The present data revealed that this disulfide bond enhanced resistance to the pull-induced CD172a dissociation from CD47 by modifying the CD47 orientation relative to the cellular membrane surface and the tensile direction along CD172a/CD47 axis under a mechano-microenvironment (Figure 4A,C,D).

The patterns of interfacial H-bonds indicated a diverse pull-induced evolution of the H-bond interaction on the binding site of WT or GT complex with one or two fixed atoms (Figure 4B). The synergistic interactions of the interfacial H-bonds contributed to resistance pull-induced dissociation of the complex. A similarity existed in the two systems for WT complex, and for GT complex (Figure 4B), although all the H-bond occupancy patterns were different from each other. However, the residue pairs, LYS96 on CD172a with GLU104 on CD47 and ARG69 on CD172a with both GLU100 and GLU35 on CD47, contributed three H-bonds with higher occupancies (Figure 4B), meaning that these residues provided a dominant resistance to the pull-induced dissociation of the WT or GT complex with or without constraint on CYS15 on CD47.

### 2.4. Catch Bond Mechanism Mediated the PTM-Enhanced CD172a-CD47 Interaction by Making the CD172a Binding Pocket Contract Further

To examine mechanical regulation of CD47 dissociation from CD172a under a physiological mechano-microenvironment, we performed “ramp-clamp” SMD simulation on each of the WT and GT molecular systems for 40 ns thrice at a tensile force of 25 pN just using loading mode 2, which was selected to model the physiological native disulfide bond from CYS245 paired with CYS15 in CD47 [17]. We found that the Cα-RMSD fluctuated in a range from 1 to 2 Å during the simulation duration (Figure 5A), while the distance from the steered atom to the fixed one stayed at a plateau with a slight roughness of height of about 4 Å (Appendix A), suggesting a stable conformation of the stretched complex with an allosteric modulation. The tensile force of 25 pN caused an increase in H_NB_ (the mean interfacial H-bond number over 40 simulation time thrice) and a slight decrease in either the dissociation probability (*f*_D_) or the binding energy (E_B_), no matter whether the CD172a-bound CD47 was WT or GT (Figure 5B–D). In spite of that, the tensile change in either Buried-SASA or in Rgyr of the complex with or without PTM was negligible (Appendix A). However, these data suggested a catch bond mechanism for the PTM-enhanced CD172a-CD47 interaction.

To uncover the structural basis of the catch bond phenomenon in interaction of CD47 with CD172a, we measured the three abovementioned geometric characteristics (*H*, *α* and *θ*) (Figure 3A–C) of the complex from the “ramp-clamp” SMD runs at static and tensile force of 25 pN. In comparison with the case at static state, we found that the tensile force of 25 pN caused a slight decrease in either the distance *H* and the angle *α* (Figure 6A,B), leading to a force-induced contraction of the CD172a binding pocket. This force-induced change in *H* and *α* was pronounced more clearly in the GT rather than the WT complex (Figure 6A,B), and the angle *θ* remained almost constant for the WT complex but had a slight increase for the GT one (Figure 6C), indicating a PTM-mediated enhancement of mechanical stabilization of the complex structure. The present data suggested that a catch bond mechanism might mediate PTM-enhanced CD172a-CD47 interaction by making the CD172a binding pocket contract further.

Additionally, we found that all the thirteen interfacial H-bonds could be clustered to three types, type I, II, and III, based on their responses to the tensile force (Figure 6D). The H-bonds in the type I were force desensitized, and consisted of two members contributed by GLU104 on CD47 with LYS96 on CD172a and GLU35 on CD47 with ARG69 on CD172a. All six force-enhanced H-bonds were assigned to type II, and contributed by GLU104 on CD47 with GLN52 and LYS53 on CD173, ASP46, PCA1, THR102, and ASP51 on CD47 with their respective partners, such as SER98, SER66, ARG69, and ARG95 on CD172a. Meanwhile, five residue pairs, consisting of GLU106, GLU100, GLU97, LYS39, and LYS36 on CD47 with their respective partners, such as LYS53, ARG69, LYS96, ASP100, and GLU54 on CD172a, contributed the other five force-weakened H-bonds in the type III (Figure 6D). However, it was the synergistic effect of the thirteen H-bonding interactions that induced the catch–bond phenomenon in CD47-CD172a interaction. Additionally, the residue ARG69 on CD172 with its partner on CD47 might be crucial to the structural stability of the complex because of their relevant H-bonds of high occupancies (Figure 6D).

## 3. Discussion

The post-translated modification (PTM) of CD47, or the conversion of glutamine (the 1st residue in CD47 *N*-terminus) to pyroglutamic acid via catalysis of glutaminyl cyclase on Golgi apparatus, was highly related to CD47 functions [17,22,23]. Meanwhile, the cells, which adhered to ECM or interacted with each other, would deform or move away from each other in response to mechanical stimulus, such as fluid shear stresses, squeezing, and tearing forces from surrounding tissues or cells [31]. Together with cell rolling or tethering, the cytoskeletal rearrangement, membrane expansion, and contraction of the deformed cells triggered transmembrane mechano-signaling, possibly leading to a remarkable tensile force on the CD47-CD172a linker between the cell and ECM or cell. Better mechanical strength and stability of the CD47-172a complex should be required for formation and stability of cell–cell cross-talking under diverse mechano-microenvironments in the presence and absence of PTM. With a series of free and steered MD simulations, we demonstrated that PTM induced enhancement of CD47 binding to CD172a through contraction of the CD172a binding pocket either at static or under loading status. Support for this statement came from the fact the *N*-terminal PTM did not affect CD47 expression on cytomembrane but enhanced binding of CD172a in solution to CD47 on membrane, and the pharmacological inhibition or absence of the glutaminyl cyclase would inhibit CD47 signaling [22,23,37].

CYS15 in extracellular domain could form a disulfide bond with CYS245 on the transmembrane domain in CD47 [17]. This disulfide bond is sensitive to changes in redox potential, providing an additional membrane anchor point to likely constrain the orientation of CD47 relative to the cell membrane. It thereby prompted CD47–ligand binding and subsequent signal transduction [14,17,18]. To model the CYS15 anchor, we fixed the two Cα-atoms on CYS15 and VAL115 of CD47 for SMD runs. The present data exhibited that the constraint on CYS15 would induce enhancement of the mechanical strength of CD172a/CD47 complex with or without *N*-terminal PTM (Figure 4D), in better consistency with the statement for the disulfide bond (between CYS245 and CYS15)-induced improvement of CD47 adhesion function [14,18]. The residue GLN52 on CD172a with its partner CLU104 on CD47 might serve as anther pivot in responding to PTM, coming from the dramatic increase in the occupancy of H-bonding between this residue pair (Figure 4B; Table 1).

Conceivably, CD47 *N*-terminal PTM would trigger a cascade of intramolecular events for the CD47/CD172a complex. Through hydrogen interaction with SER66 on CD172a, the pyroglutamic acid on *N*-terminus of CD47 triggered remodeling of the interfacial H-bond occupancy pattern of the complex (Table 1) first, then caused an increase in the interfacial surface and a change in interfacial topology (Figure 2E and Figure 3). It thereby enhanced binding of CD47 with CD172a. Being highly relevant to remodeling of interfacial H-bonding and binding site, the dynamics-driven and/or tensile force-induced contraction of the CD172a binding pocket with better flexibility was a crucial event in the cascade mentioned above (Figure 3). Additionally, we found that the stretched complex was stable under the given tensile force of 25 pN, and the force would enhance binding of CD172a to CD47 with or without PTM, but the complex with PTM had better mechanical stability in comparison with one without PTM (Figure 5C). This catch–bond phenomenon might be required for a stable cell–cell crosstalk, has been observed in various molecular systems, including P-selectin bound with PSGL-1 using the force clamp assay with AFM and a flow chamber [38], as well as Kindlin2 bound with β3 integrin and Mac-1 bound with GPIbα using molecular dynamics simulations [39,40]. So, the aforementioned data from SMD simulations stated that CD47 was mechano-sensitive, had a stable tension-induced allostery with an enhanced affinity to CD172a, and served not only linker molecules but also as a mechano-chemical signaling axis in cell–cell cross talking through binding with CD172a, while PTM raised the mechanical strength of the CD47-CD172a axis. The residue ARG69 on CD172a with its partner on CD47 might be crucial to the structural stability of the complex (Figure 6D), because of their relevant H-bonds of high occupancies. However, the effects of *N*-terminal PTM on CD47/CD172a complex dissociation were more significant than that of tensile force.

In conclusion, we determined from the MD simulation that both *N*-terminal PTM and tensile force did regulate binding of CD47 and CD172a. Our results suggested that the *N*-terminal PTM of CD47 mediated the dynamics-driven interfacial H-bonding enhancement of the complex, along with contracting of the binding pocket in the bound CD172a. An additional constraint on CYS15 on CD47 significantly improved the tensile strength of the complex with or without PTM, and a catch–bond phenomenon would also happen in complex dissociation under tensile force of 25 pN in a PTM-independent manner. Additionally, several binding site residues, such as GLN52, SER66, and ARG95 on CD172a, were qualified as candidates of the key residues in the PTM-enhanced binding of CD172a to CD47. This MD simulation prediction provided an explanation for the mechano-chemical regulation on the CD47 binding with CD172a at atom level, which might serve as the molecular basis for the PTM-induced function improvement of CD47. It would also be helpful for a deeper understanding CD47-relevant cellular immune response and cancer development and for developing new strategies of immunotherapy targeting this adhesion molecule interaction.

## 4. Materials and Methods

### 4.1. System Setup

Two molecular systems were set up using the CD47/CD172a complexes of wild-type (WT) and GLN-type (GT). A post-translational modification occurred on the *N*-terminal of CD47 in the WT complex and did not in the GT one. The crystal structure of WT CD47 (the 1st–115th residue)/CD172a (the 2nd–115th) complex was read from the Protein Data Bank (PDB ID: 2JJT) (Figure 1) [17], while the initial structural model of the GT complex was created by replacing residues 1–3 in the WT complex with residues 1–3 of CD47 bound with antibody B6H12 (PDB ID:5TZU) [41]. The initial model of the GT complex was energy minimized along the protocol, such that all atoms except either the third–fourth residues of CD47 at the first 1000 minimization steps or the first–fourth residues of CD47 at the followed 2000 minimization steps were fixed, and the last 5000 minimization steps were run for all the unconstrained atoms [42]. Each of the WT and GT complex structures was solvated in TIP3P water in a rectangular box with walls at least 15 Å away from any protein atom using VMD 1.9.2 [43]. The system was then neutralized with 150 mM NaCl to resemble the physiological environment.

### 4.2. Molecular Dynamics Simulation

Two software packages, VMD 1.9.2 and ChimeraX 1.4, were used for visualization and modeling [44], while MD simulations were performed using NAMD 2.13 with CHARMM36 force field [45]. Each system was subjected to an energy minimization first and then equilibrated thrice for 50 ns. A stable complex structure in each equilibrated system was chosen as the initial conformation for production simulations for each system. Free MD simulation was run thrice on each system over 40 ns.

The so-called “ramp-clamp” SMD simulations, a force-clamp MD simulation followed by a force-ramp one, were performed on each equilibrated system to examine the force-induced changes in complex conformation and function in the presence or absence of the post-translational modification of CD47. Two loading modes, mode 1 and mode 2, were applied in SMD simulations for studying mechanical constraint effects on CD47 function. In mode 1, just the Cα-atom of VAL115 on CD47 was fixed, and the steered atom, the C-terminal Cα atom of CD172a (ALA115), was pulled along a direction from the fixed atom to the pull atom. In mode 2, two Cα-atoms of CD47, one on VAL115 and another on CYS15, were fixed, and the Cα atom of CD172a C-terminal (ALA115) was the steered one, but pulling took place along the direction from the center of the line between two fixed atoms to the steered atom. For the loading manner in mode 2, we fixed CYS15 in the extracellular domain of CD47 to form a disulfide bond with CYS245 in its transmembrane domain [17]. Model 2 was used to model the disulfide bond between CYS15 and CYS245, which lay in the extracellular and transmembrane domain of CD47, respectively [17]. The virtual spring connecting the dummy atom and the steered atom had a spring constant of 34.74 pN/nm. For each system at either loading mode 1 or 2, the force-camp MD simulation ran over 10 ns thrice with a time step of 2 fs and a constant velocity of 5 Å/ns, at which the pulling would be contributed to complex dissociation with secondary structure conservation.

Once tensile force reached 25 pN, the SMD simulation was transformed from the force-ramp mode to a force-clamp one, at which the complex was stretched with the given constant tensile force for the following 40 ns thrice. Herein, only loading mode 2 with two fixed CD47 Cα-atoms (one on VAL115 and another on CYS15) was applied to the force-clamp simulations, for simplicity. Each event of hydrogen bonding under stretching was recorded to examine the involved residues and their functions.

### 4.3. Data Analysis

VMD tools were used to analyze atom trajectories from MD simulations. The Cα-root mean square deviation (RMSD), the solvent accessible surface area (SASA, with a 1.4 Å probe radius) and the rotation radius (Rgyr) of complex were measured to estimate the structural stability, the complex interface geometry characteristics, and the tightness of complex structure, respectively. The number of hydrogen bonds (H-bonds) on the complex interface was calculated with a cutoff donor–acceptor distance of 3.5 Å and a cutoff donor–hydrogen–acceptor angle of 30°. The occupancy (or survival ratio) of a H-bond was evaluated based on the fraction of bond survival time in the simulation period. Additionally, the probability of ligand dissociation from receptor was calculated to approximately evaluate the receptor’s affinity to its ligand, as described in our previous works [39]. The binding energy consisted of gas-phase energy difference between the complex and the separated receptor and ligand, the polar solvation free energy, and the nonpolar solvation-free energy.

### 4.4. Statistical Analysis

Statistical analysis was performed using GraphPad Prism version 9.0.0 (GraphPad Software, Inc., La Jolla, CA, USA). Unpaired two-tailed Student’s t-tests were used to determine the statistical significance of the differences between the data obtained from the WT and GT systems (* *p* < 0.05; ** *p* < 0.01; *** *p* < 0.001). Two-way analysis of variance (ANOVA) with Tukey’s multiple comparisons was employed to assess the statistical significance of the data obtained from the SMD simulations (* *p* < 0.05; ** *p* < 0.01; *** *p* < 0.001). All error bars represent the mean plus or minus standard deviation of the mean based on several independent experiments.

## Figures and Tables

**Figure 1 molecules-28-04224-f001:**
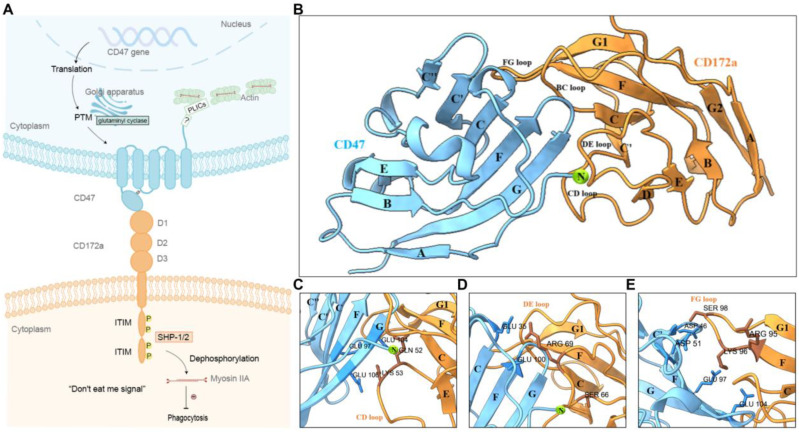
The signaling pathway, crystal structure and binding site of the CD47/CD172a complex. (**A**) The schematic diagram for CD47-CD172a axis. Interaction of CD47 and CD172a induces the phosphorylation of two tyrosine residues within the immunoreceptor tyrosine-based inhibition motif (ITIM) of SIRPα’s cytoplasmic domain. This event subsequently initiates the recruitment and activation of SHP1 and SHP2, leading to a signaling cascade that culminates in the dephosphorylation of myosin IIA and resulting in the inhibition of phagocytosis. Upregulation of CD47 expression is one of the mechanisms underlying the increased activity of the CD47-CD172a axis, while the post-translational modification of CD47 is involved in this aforementioned process, too. (**B**) The crystal structure of the complex of CD47 with CD172a. The crystal structure of the complex (PDB ID: 2JJT) is shown using a new cartoon representation. The CD47 extracellular IgV-like domain (in cyan) (the 1st–115th residue) is responsible for binding with CD172a. It contains three short β-strands (A, B and E), two helices, and five long β-strands (C, C′, C″, F and G) [17]. The first extracellular Ig subdomain (the 2nd–115th) (in orange) of CD172a binds to CD47, is consisting of eight β-strands (A, B, C, C″, D, E, F, G1, G2 and F) and their linker loops. The Cα-atom of pyroglutamic acid (green) at position 1 in the residue sequence of the wild-type (WT) CD47 is represented in van der Waals mode. The crystal structure of CD47 (in cyan) bound to CD172a (in orange) is shown in a new cartoon representation. (**C**–**E**) The binding site of CD47/CD172a complex. The labeled residues in CD47 or in the CD loop (**C**), DE loop (**D**) and FG loop (**E**) of CD172a were involved in binding to CD47.

**Figure 2 molecules-28-04224-f002:**
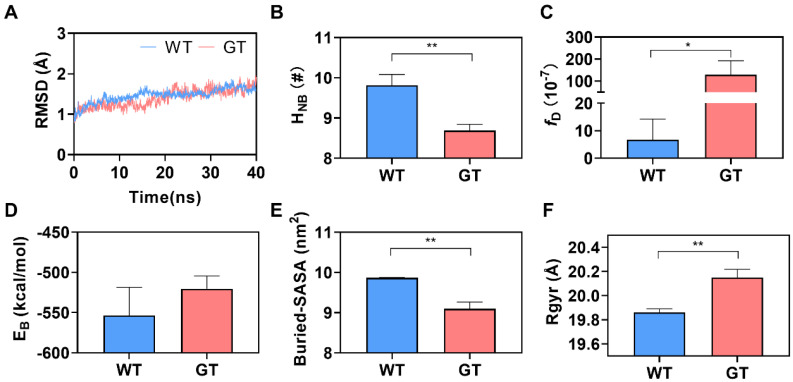
Effect of *N*-terminal modification on interaction of CD47 with CD172a. (**A**) The mean time-courses of Cα-RMSD, (**B**) the mean interfacial H-bond number (N_HB_), (**C**) the dissociation probabilities (*f*_D_), (**D**) the mean binding energy (E_B_), (**E**) the mean buried SASA, and (**F**) the mean rotation radius (Rgyr) of either WT or GT CD47/CD172a complex over 40 ns production simulation. The *p*-values of the unpaired two-tailed Student’s t test were shown to indicate the statistical difference significance (* *p* < 0.05; ** *p* < 0.01), or lack thereof. Data were shown with mean ± S.D of three runs.

**Figure 3 molecules-28-04224-f003:**
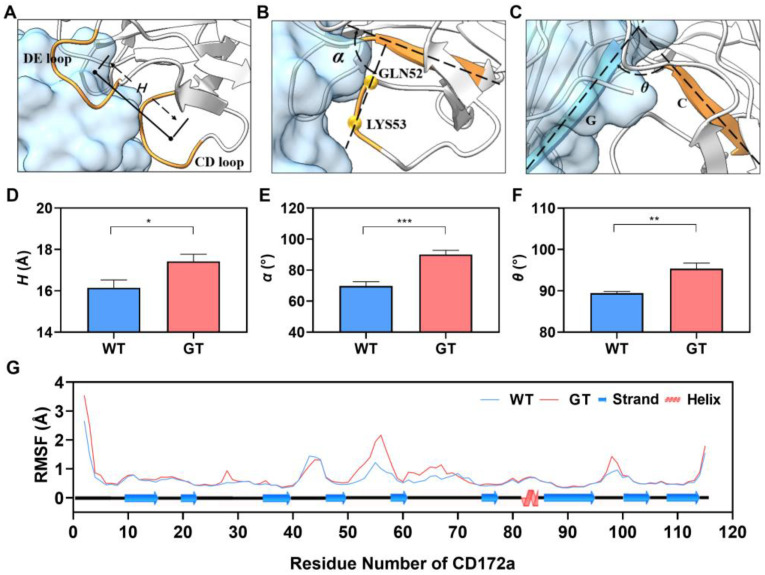
PTM-induced Allostery of the bound CD172a. The schematic diagram for (**A**) the distance *H* between the centroids of CD loop and DE loop in CD172a, (**B**) the angle *α* between the C strand in CD172a and the straight line extending from the Cα-atom of GLN52 to the Cα atom of LYS53 in CD172a, and (**C**) the angle *θ* between the G strand in CD47 and the C strand in CD172a. The plots of the distance *H* (**D**), the angle *α* (**E**), and *θ* (**F**) over thrice 40 ns simulations. (**G**) The mean RMSF of the bound CD172a over three 40 ns simulation. WT or GT denoted the bound CD47 with or without PTM, respectively. The *p*-values of the unpaired two-tailed Student’s t test were shown to indicate the statistical difference significance (* *p* < 0.05; ** *p* < 0.01; *** *p* < 0.001), or lack thereof. Data were shown with mean ± S.D of three runs.

**Figure 4 molecules-28-04224-f004:**
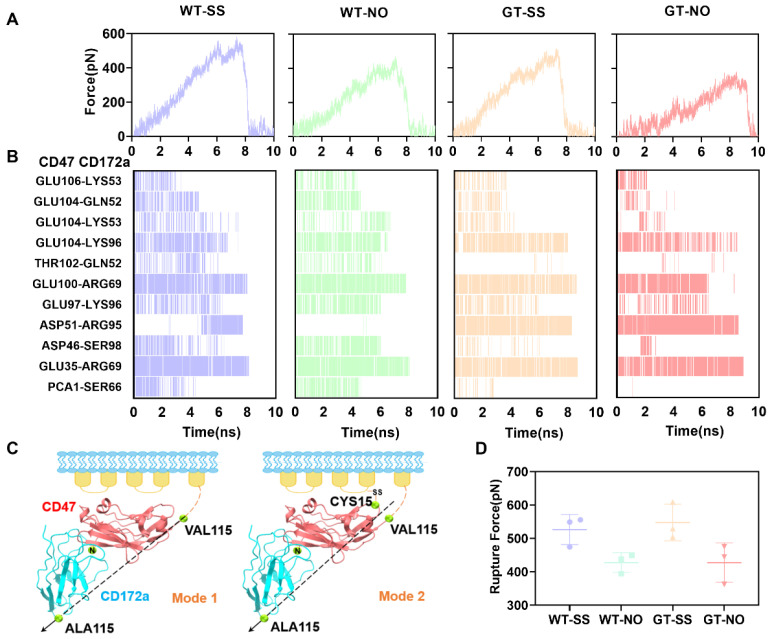
Pull-induced dissociation of CD172a from CD47. The data were from “force-ramp” SMD runs over thrice 10ns using pulling velocity of 5 Å/ns along direction from steered Cα atom on ALA115 on CD172a to either the Cα atom on VAL115 of CD47 or the center of the line between two fixed Cα atoms on VAL115 and CYS15 of CD47. (**A**) The representative time curves of loading force for four different systems, the WT or GT complex with one or two fixed Cα atom. The symbols, WT-SS and GT-SS, expressed the WT and GT complexes, in which the two Cα atoms on VAL115 and CYS15 of CD47 were fixed, respectively. The symbols, WT-NO and GLN-NO, were assigned to the WT and GT complexes, in which only the Cα atom on VAL115 of CD47 was fixed, respectively. (**B**) The occupancy patterns of interfacial H-bonds for a representative for each system. (**C**) Schematic diagram of the ligated CD47 under stretching. (**D**) The scatter plot of the rupture force for each system.

**Figure 5 molecules-28-04224-f005:**
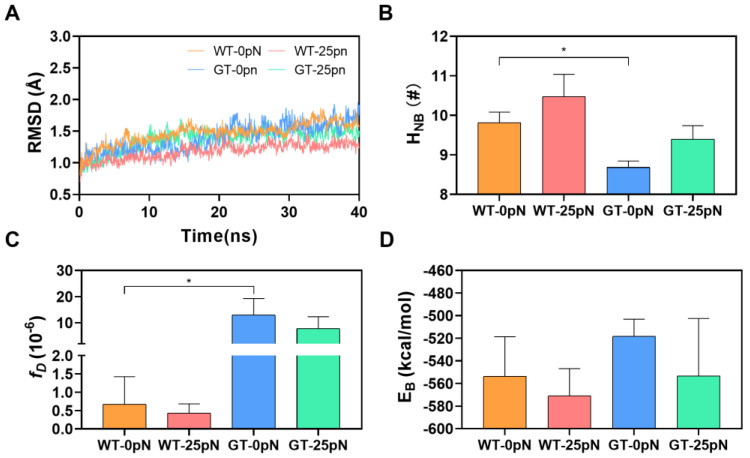
Interaction of CD47 with CD172a under tensile force of 25 pN. All data were sampled from “force-clamp” SMD runs of 40 ns thrice for WT-SS and GT-SS system at static state or under tensile force of 25 pN. (**A**) The representative time-courses of Cα-RMSD, (**B**) the mean hydrogen bond number (H_NB_), (**C**) the dissociation probability (*f*_D_), and (**D**) the mean binding energy of WT or GT CD47/CD172a complex at static state or under tensile force of 25 pN. The *p*-values of two-way analysis of variance (ANOVA) with Tukey’s multiple comparisons were shown to indicate the statistical difference significance (* *p* < 0.05), or lack thereof. Data were shown with mean ± S.D of three runs.

**Figure 6 molecules-28-04224-f006:**
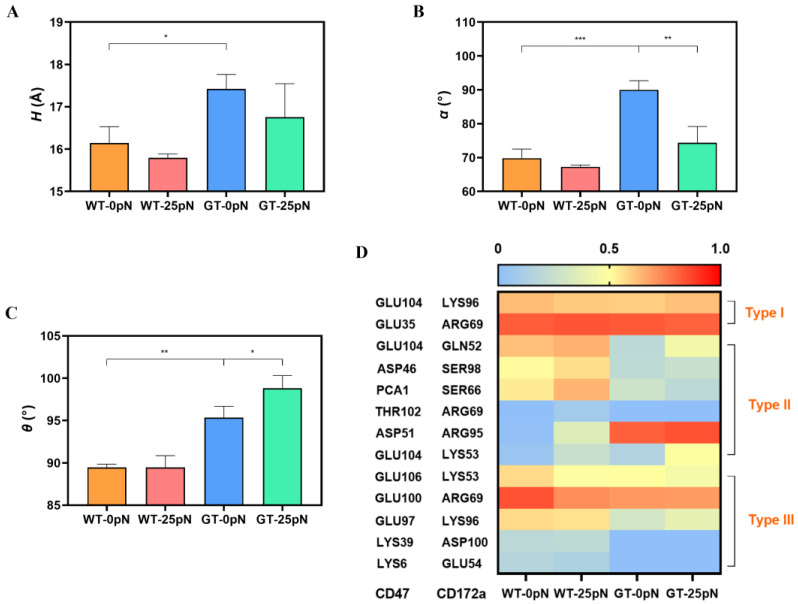
Tension-induced structural change in CD172a and the involved residue pairs at binding site. The plots of the distance *H* (**A**), the angle *α* (**B**), and *θ* (**C**) over thrice 40 ns “force-clamp” SMD runs for four systems, the WT-SS and GT SS systems at static state or under tensile force of 25 pN. (**D**) The heatmap of H-bond occupancies and their involved residue pairs under different conditions. The *p*-values of two-way analysis of variance (ANOVA) with Tukey’s multiple comparisons were shown to indicate the statistical difference significance (* *p* < 0.05; ** *p* < 0.01; *** *p* < 0.001), or lack thereof. Data were shown with mean ± S.D of three runs.

**Table 1 molecules-28-04224-t001:** Key residues pairs in binding of CD172a to CD47 with or without PTM.

Number	Residue		Occupancy	
	CD47	CD172a	WT	GT
1	GLU106	LYS53	0.54 ± 0.04	0.48 ± 0.03
2	GLU104	LYS96	0.66 ± 0.04	0.52 ± 0.10
3	GLU104	GLN52	0.60 ± 0.03	0.22 ± 0.07
4	GLU100	ARG69	0.79 ± 0.06	0.69 ± 0.01
5	GLU97	LYS96	0.66 ± 0.09	0.30 ± 0.09
6	GLU97	LYS53	0.34 ± 0.01	0.28 ± 0.04
7	ASP51	ARG95	0.07 ± 0.06	0.77 ± 0.07
8	ASP46	SER98	0.53 ± 0.06	0.30 ± 0.13
9	GLU35	ARG69	0.82 ± 0.01	0.81 ± 0.02
10	PCA1 or GLN1	SER66	0.52 ± 0.04	0.21 ± 0.08
11	PCA1 or GLN1	GLU54	0	0.08 ± 0.03
12	PCA1 or GLN1	GLN52	0	0.15 ± 0.04

## Data Availability

The data are available on request from the author.

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
