# Peer review of "A MD Simulation Prediction for Regulation of N-Terminal Modification on Binding of CD47 to CD172a in a Force-Dependent Manner"

_molecules, 2023, doi:10.3390/molecules28104224_

Round 1

Reviewer 1 Report

The paper from Zhao et al "A MD Simulation Prediction for Regulation of N-terminal Modification on Binding of CD47 to CD172a in a force-dependent manner"

The paper is interesting and has huge potential. There are certain issues which can be resolved to increase the scientific merit.

1. Please justify the need to perform steered molecular dynamics and regular MD simulation without applying any force

2. Request authors to provide more clarity on why tensile strength is required in presence or absence PTM is required for this study.

3. 40ns MD time is very less for interfering the results. I request authors to run for at least 100ns which is a standard bare minimum.

4. Please provide justification for only 10ns SMD?

5. Which tool or software has been used for statistical analysis?

Reviewer 2 Report

The authors already involved in similar type of research which revealed their interest in the area of research as they recently published in similar journal Fang, L.; Zhao, Y.; Guo, P.; Fang, Y.; Wu, J. MD Simulation Reveals Regulation of Mechanical Force and Extracellular Domain 2 on Binding of DNAM-1 to CD155. Molecules 202328, 2847.

For me I think article is expressing good data related the title. This article might be part of journal after some minor corrections.

N should be italic for N-terminal and for all the letters representing amino group in all the text.

Similarly, symbols like theta alpha etc. should be italic.

Article should be read carefully and improve for any errors.

Author Response

Comment 1. N should be italic for N-terminal and for all the letters representing amino group in all the text. Similarly, symbols like theta alpha etc. should be italic.

Answer: We are sorry for these mistakes. We have corrected them in the revised manuscript.

Reviewer 3 Report

This report study the interaction of: CD47 (cluster) NTD with the SIRPα protein and the authors claim they have provided "an explanation for the mechano-chemical regulation on the CD47 binding with CD172a at atom level".

First thing, this work fits better in a more "biological" journal as there is very little chemistry here (even chemical biology). This is a manuscript more appropiate for Biomolecules or Polymers, both within MDPI editorial.

The MD simulations (free and constrained) used here are standar and routine methods and they seem to be well performed. So, there is not originality based on the methods. There could be an original work in terms of the system but I have some concerns about the relaibility of the model. 

CD47 is a big cluster of prooteins, interacting with SIRPα that is another conglomerate of proteins, both inmersed in different membranes. This model seems to me an oversimplication of the system (Maybe this is why is submitted to a chemistry journal instead of a biochemistry/biology one?).

In my opinion it is mandatory to add an introductory figure showing the size and extend of the big system (membranes on tummor cell, macrophage, different domains of the clusters of proteins). This will give the reader an idea of the simplication of the model and the studied interactions. 

The authors should justify beter how they think this small system may explian the "mechano-chemical regulation" of such a big conglomerate of interating proteins. 

If this was a totally chemicaly-driven process I would say the model could be appropriate as the effect could be more local but the effect studied is mechanical so a better explanation of how the whole system affect (or not) the local interactions is desirable. 

Round 2

Reviewer 1 Report

None

Reviewer 3 Report

I still think this is a too complex system to draw such an ambitious conclusion but, as the author said in their response: "It is difficult to obtain a rational result from the complicated system within limited computational resources". So, if the model is accepted as valid then, the work done is solid and sound and have some value. I would say it is publishable in its current form.